# Biological Effects and Biomedical Applications of Areca Nut and Its Extract

**DOI:** 10.3390/ph17020228

**Published:** 2024-02-08

**Authors:** Ting Tong, Aiqing Xu, Shuhua Tan, Hengzhi Jiang, Lixin Liu, Senwen Deng, Haihua Wang

**Affiliations:** Hunan Key Laboratory of Economic Crops Genetic Improvement and Integrated Utilization, Hunan Binglang Science Institute, School of Life and Health Sciences, Hunan University of Science and Technology, Xiangtan 411201, China; aqxu@hnust.edu.cn (A.X.); shtan@hnust.edu.cn (S.T.); m18874674865@163.com (H.J.); m17378201408@163.com (L.L.); dswwzls@hnust.edu.cn (S.D.)

**Keywords:** areca nut, betel nut, anti-inflammatory activity, antioxidant activity, antibacterial activity, antidepressant activity, toxicity

## Abstract

The dried, mature fruit of the palm tree species *Areca catechu* L. is known as the areca nut (AN) or betel nut. It is widely cultivated in the tropical regions. In many nations, AN is utilized for traditional herbal treatments or social activities. AN has historically been used to address various health issues, such as diarrhea, arthritis, dyspepsia, malaria, and so on. In this review, we have conducted a comprehensive summary of the biological effects and biomedical applications of AN and its extracts. Initially, we provided an overview of the constituents in AN extract. Subsequently, we summarized the biological effects of AN and its extracts on the digestive system, nervous system, and circulatory system. And we elucidated the contributions of AN and its extracts in antidepressant, anti-inflammatory, antioxidant, and antibacterial applications. Finally, we have discussed the challenges and future perspectives regarding the utilization of AN and its extracts as emerging pharmaceuticals or valuable adjuncts within the pharmaceutical field.

## 1. Introduction

*Areca catechu* L., known as the areca nut palm, is an evergreen tree belonging to the genus *Areca* of the palm family (Arecaceae). It is native to Malaysia and primarily distributed in tropical regions of Asia, Africa, and Europe. Areca nut (abbreviated as AN), also referred to as betel nut, is the ripe fruit of *Areca catechu* L. and is widely used as an essential component in many traditional Chinese medicinal formulations [1]. Figure 1 shows the typical anatomical structure of mature areca nuts (abbreviated as ANs), which are prominently composed of kernels and pericarps [2]. In traditional Chinese medicine, kernels are commonly utilized after slicing and drying and are referred to as semen arecae. Pericarps are used as well after cooking, loosening, and drying and are referred to as pericarpium arecae [3]. AN is also processed for traditional snack purposes. In this case, AN is often called betel quid. In Southeast Asia, fresh betel nuts are commonly chewed with slaked lime and Laotian leaves or grated tobaccos [4]. In China, dried betel nut (dried pericarp) is usually processed into refined betel nuts, referred to as “Binglang” by the local consumers, through complex processing procedures involving boiling, flavor addition, brine addition, and drying [5].

Areca nut extracts (abbreviated as ANEs) were usually obtained through the solvent extraction method, including impregnation, percolation, decoction, reflux extraction, ultrasonic extraction, microwave extraction, etc. With advancements in science and technology and in-depth research by scholars, various physiologically active ingredients such as areca alkaloids, polyphenols, tannins, amino acids, fatty acids, polysaccharides, saponins, and others were isolated. Fifty-nine compounds have been identified using ultra-high-performance liquid chromatography–tandem mass spectrometry and spectral analysis technology [1].

AN is widely utilized in various regions, including China, India, and other countries in South and Southeast Asia for its medicinal value [3,6]. The use of betel nut and its extracts as traditional Chinese medicine can be traced back to 1953 in the Chinese Handbook of Pharmacology [7], which includes 60 preparations related to AN. These preparations are commonly employed to address digestive and endocrine disorders [8]. In recent years, with advancements in science and technology, researchers have analyzed the biological effects of ANs and ANEs, discovering their effects on the digestive system, nervous system, blood circulation system, and so on. To date, ANs and ANEs have been utilized as insect repellent, anti-thrombotics, anti-atherogenics, anti-inflammatories, antibacterials, antivirals, and antidepressants [9,10,11]. Meanwhile, several review articles have examined other aspects related to the biological effect of AN, including its potential carcinogenic effects [12], addictive properties [13], association with oral cancer [14,15,16,17,18], carcinogenesis [19,20], and liver disease [21].

In this review, we summarized the traditional uses, biological effects, and biological applications of ANs and ANEs. First, we classified and summarized the ANEs. Subsequently, we summarized the biological effects of ANs and ANEs on the digestive system, nervous system, and blood circulatory system. Additionally, we summarize the potential of ANs and ANEs for biological applications, mainly encompassing antidepressant, anti-inflammatory, antioxidant, and antimicrobial properties. Finally, we discussed the challenges and future prospects associated with the application of ANs and ANEs as emerging drugs or valuable adjuvants in the pharmaceutical field.

## 2. Chemical Composition

AN contain a diverse range of natural bioactive compounds. To date, more than fifty-nine compounds have been isolated and identified from AN. The primary chemical constituents found in ANs include alkaloids, flavonoids, tannins, terpenoids, steroids, fatty acids, and amino acids. Furthermore, ANs also contain polysaccharides, pigments, and saponins, along with other compounds (Table 1).

Previous articles have extensively discussed the identification and chemical structures of these substances derived from ANs [1], and therefore, we will not reiterate their findings in this review. We have listed the main bioactive phytoconstituents of AN in Figure 2.

## 3. Biological Effects

Recent pharmacological investigations have demonstrated that AN and ANE exhibit a diverse array of pharmacological effects. These effects encompass the facilitation of digestion, manifestation of antidepressant properties, demonstration of anti-inflammatory effects, display of antioxidant activity, possession of antiparasitic attributes, and manifestation of antibacterial effects [1,27].

### 3.1. The Effects of Areca Nut on the Digestive System

The main alkaloid in AN, arecoline, functions as an agonist on muscarinic acetylcholine receptors, leading to cholinergic effects in the parasympathetic nervous system [28,29]. Previous investigations have confirmed that arecoline can activate calcium channels in colonic smooth muscle cells, leading to muscle contractility through the engagement of M2 receptors and the initiation of protein kinase activity [30]. This finding substantiates the potential role of arecoline in the modulation of the gastrointestinal system. In an *in vitro* study conducted by Xie et al. [31], it was observed that the application of different concentrations of arecoline to isolated muscle strips resulted in a contraction reaction within 3 min. This contraction reaction could be blocked by atropine, indicating that arecoline promotes intestinal peristalsis by acting on M3 receptors. Li et al. also demonstrated that arecoline has a contractile effect on isolated rat distal colon smooth muscle strips, with the contractile response increasing with higher dosages [32]. Furthermore, arecoline supplementation has demonstrated an enhancement in the digestive and absorptive capacity in fish, potentially mediated by the up-regulation of M3 mRNA levels and substance P content. This effect appears to be closely associated with the mitigation of oxidative damage, possibly attributed to heightened activation of the Keap1a/Nrf2 signaling pathway. Additionally, it has been suggested that the rise in apical junction complex and the suppression of the RhoA/ROCK signaling pathway may be linked to the effects of arecoline. Based on a comprehensive assessment of various growth indicators, lipase activity, intestinal mucosal permeability, and antioxidant indexes, specific recommended dosages of arecoline supplementation for adult grass carp (608–1512 g) were determined. These dosages are as follows: 1.20, 1.21, 1.07, and 1.19 mg/kg, respectively [33]. Additionally, alkaloids have demonstrated parasiticidal properties against various parasites, including ascaris lumbricoides, hookworm, fasciola hepatica, and small melon worms. For instance, arecoline has been observed to depolarize the membrane potential of Ascaris lumbricoides muscle cells, leading to their depolarization. Jain et al. established the lethal impact of arecoline hydrobromide on cattle tick Rhipicephalus (Boophilus) microplus. Larval packet tests indicated that a concentration of 12.5 mg/mL of arecoline resulted in 100% larval mortality [34]. Furthermore, adult immersion tests revealed that arecoline led to an 87.90% mortality rate in ticks, reduced egg weight, and a significant decrease in hatchability. These outcomes were linked to a reduction in the activities of MAO and AChE enzymes, as well as diminished levels of protective enzymes, including SOD and GST. As a result, arecoline demonstrates robust acaricidal properties and the ability to induce oxidative stress in bovine ticks [35].

In a study conducted by Zhou et al. and colleagues, the effects of integrating AN on the gut microbiota of mice subjected to a high-fat diet were investigated. This result revealed that AN successfully reduced body weight and serum lipid profiles in mice following a high-fat diet. Furthermore, the gut microbiota composition was modulated, leading to the prevention of adipose tissue accretion [36]. Furthermore, studies have demonstrated that the co-administration of ANE and pumpkin seed can be employed for the management of human tapeworm infection, yielding respective cure rates of 63.6% and 75%. When utilized in combination, these two agents manifest a synergistic effect, resulting in a more effective therapeutic outcome. The average time for worm excretion was found to be only 2 h, with a cure rate of 88.9% [37].

Another study assessed the influence of AN on gastrointestinal tract movement and the levels of gastric inhibitory polypeptide, glucagon-like peptide-1, gastric juice, and bile in rat subjects. The alkaloid levels exhibited a swift decline post-treatment. Nonetheless, the enhancement of gastrointestinal motility remained unabated [6]. Tey et al. investigated the effects of extracts from the pericarpium of the AN and arecoline on the contraction of porcine lower esophageal sphincter sling and clasp muscles. It was found that the contraction of the lower esophageal sphincter caused by these extracts and arecoline was not affected by tetrodotoxin inhibition but was significantly suppressed by atropine. This suggests that the contractions of the porcine lower esophageal sphincter induced by the extracts and arecoline of the AN may be dose-dependent and mediated through muscarinic receptors [38].

The study conducted by Wang et al. [39] provides evidence that arecoline has a direct impact on the Leydig cells in the testes, resulting in an elevation in testosterone secretion. The researchers also highlight that this mechanism is achieved by activating L-type calcium channels, which subsequently enhance the activity of 17β-hydroxysteroid dehydrogenase and the expression of steroidogenic acute regulatory (STAR) protein. Furthermore, other investigations suggest that arecoline can stimulate the release of endogenous adrenocorticotropic hormone by activating the hypothalamic–pituitary–adrenal axis [8]. In a separate study by Lim et al. [40], it was found that arecoline, through the activation of cholinergic receptors, effectively inhibits the release of catecholamines from the isolated adrenal glands of rats in a dose-dependent manner.

### 3.2. The Effects of Areca Nut on the Nervous Systems

Arecoline possesses the ability to stimulate the parasympathetic nervous system, exhibiting a mechanism of action similar to that of acetylcholine. Additionally, it exerts central inhibitory effects. Intraperitoneal administration of arecoline has been observed to counteract the behavioral excitement induced by pilocarpine and scopolamine, as well as antagonize brain-wave dissociation. At specific dosages, arecoline can stimulate M receptors to compensate for acetylcholine deficiency, thereby promoting bodily excitement and enhancing learning and memory capabilities. In the presence of slaked lime, arecoline and arecaidine undergo hydrolysis into arecaidine and arecadine, respectively. Both hydrolyzed products can bind to γ-aminobutyric acid (GABA) receptors in the brain, exerting inhibitory physiological activity. Consequently, they prevent the inhibitory effects of GABA on neural transmission, leading to pleasurable sensations.

Research conducted by Sun et al. [41] demonstrated that systemic administration of arecoline to rats can diminish ethanol-induced sedation by activating M receptors in the central nervous system. Extensive experimental investigations consistently reveal that arecoline, functioning as a muscarinic receptor agonist, readily crosses the blood–brain barrier and stimulates central M receptors. Furthermore, evidence suggests that scopolamine, acting as a central muscarinic receptor antagonist, can counteract the inhibitory effects of arecoline on autonomic activity. However, this inhibitory effect cannot be counteracted by another muscarinic receptor antagonist, methylscopolamine, or by nicotine receptor (N receptor) antagonists that are incapable of crossing the blood–brain barrier [42].

Chandra et al. [43] conducted an investigation employing Wistar rat models of Alzheimer’s disease (AD) and observed that the administration of arecoline and thioacetamide elicited an augmentation in neuronal memory cell count and enhancement in cognitive learning capabilities in the rats. Similarly, Maiese et al. [44] reported that clinical administration of arecoline exhibited the potential to augment cognitive capacities and memory in patients with AD. Additionally, *in vitro* experiments have provided evidence of arecoline’s ability to facilitate recuperation from profound peripheral nerve injuries [45]. Furthermore, Michiko et al. [46] conducted experiments on animal models and found that arecoline, acting as a cholinergic pharmacological agent, could alleviate age-related impairment of time perception in rats, thereby improving symptoms in AD patients. Epidemiological investigations have also indicated a significant association between the symptoms exhibited by schizophrenia patients in Nepal and their consumption of AN [47].

Bozorgi conducted a study to investigate the potential anti-Alzheimer’s activity of the aqueous extract of Areca catechu. Various assays were employed in the study, including a self-induced amyloid β (Aβ) aggregation assay, a beta-secretase 1 (BACE1) enzymatic assay, and a neuroprotection assay *in vitro*. The results demonstrated that the extract exhibited inhibitory activity against the accumulation of Aβ and BACE1, as well as neuroprotective effects [48]. Furthermore, the extract was found to reverse scopolamine-induced memory deficits in rats, as evidenced by the Morris water maze test conducted *in vivo*. Specifically, concentrations of 1.5 and 3 mg/kg of the extract were observed to enhance memory performance. Another study by a different researcher investigated the effects of the aqueous extract of *Piper betle* L. leaf and *Areca catechu* L. nut. This study revealed that the extract protected against pentylenetetrazole-induced seizures and positively modulated cognitive function in adult zebrafish [49].

### 3.3. The Effects of Areca Nut on Blood Circulatory System

Overnutrition has been found to have a significant impact on the prevalence of nutrition-related diseases, particularly cardiovascular diseases. Interventions such as dietary modifications or the use of pharmacological agents to target plasma triglyceride levels have demonstrated a reduction in mortality rates associated with coronary artery disease and a decrease in the overall incidence of hyperlipidemia [50]. Extracts derived from AN have been shown to inhibit the absorption of cholesterol in the body. Additionally, ethanol extract from AN has been found to reduce plasma cholesterol levels by 25% in rats fed a high-fat diet, leading to a significant reduction in the activity of intestine pancreatic cholesterol esterase, and acyl-CoA cholesterol acyltransferase A [51]. Arecoline, a compound found in these extracts, has been shown to protect against vascular endothelial damage induced by oxidized low-density lipoprotein (ox-LDL) and inhibit the excessive expression of adhesion molecules in vascular endothelial cells stimulated by high levels of fat or sugar. Furthermore, arecoline has been found to selectively act on endothelial cells, inducing vascular relaxation that is dependent on the endothelium. In animal models of early stage and mid- to late-stage atherosclerosis induced by high cholesterol, arecoline has demonstrated a protective effect in both rats and rabbits. Recent research suggests that treatment of human polymorphonuclear leukocytes (PMNs) with arecoline results in a decrease in intracellular reactive oxygen species (ROS) and myeloperoxidase (MPO) levels [52].

The water extract of AN has been identified as having a significant influence on the absorption and blood lipid content of triglycerides. This effect is attributed to the presence of proanthocyanidins in AN, which have been shown to inhibit gluconeogenesis in liver cells [53]. Byun et al. conducted a rat-based investigation, which revealed that subjects administered a combination of triglycerides and ANE exhibited reduced blood lipid concentrations in contrast to the control group. Additionally, the supplementation of ANE was found to reduce the activities of intestinal pancreatic cholesterol esterase and intestinal cholesterol acyltransferase. This suggests that the inhibition of pancreatic cholesterol esterase might lead to the suppression of pancreatic lipase activity, resulting in diminished plasma triglyceride levels [53]. Another investigation conducted by Jeon et al. [54] revealed that polyphenolic compounds derived from AN can markedly reduce blood lipid levels and the absorption of triglycerides. Furthermore, aqueous extracts from AN have been shown to exhibit anti-inflammatory and hepatoprotective effects against liver injury induced by carbon tetrachloride (CCl_4_) in rats, as evidenced by reduced levels of serum alanine aminotransferase (ALT), aspartate aminotransferase (AST), and malondialdehyde [55].

### 3.4. The Side Effects of Areca Nut

As a traditional medicine, AN has garnered significant attention due to its potential therapeutic value. However, improper usage may lead to various adverse effects. Primarily, the presence of bioactive compounds such as arecoline in AN imparts stimulant properties; excessive use has the potential to lead to addiction, compromising the overall quality of life of individuals [56,57]. Furthermore, the act of AN chewing has been linked to the occurrence of cancers, including oral and esophageal cancers, posing a potential carcinogenic risk [58,59,60]. In terms of oral health, the deposition of AN residue may result in dental discoloration and gingival inflammation, adversely affecting oral well-being [61]. Additionally, AN use is associated with cardiovascular, digestive, and neurological issues, potentially triggering conditions such as cardiovascular diseases, digestive disturbances, and neurological imbalances [61]. Prudent consideration is required, particularly in the usage of AN among adolescents and pregnant women [62]. In summary, while AN exhibits certain medicinal potential, caution must be exercised to mitigate potential risks [3]. Implementing appropriate preventive measures ensures maximal safety and health benefits. When formulating guidelines for AN usage, a comprehensive approach considering both its medicinal value and potential hazards is essential.

## 4. Biomedical Applications

Among the Arecaceae family, *Areca catechu* L. undeniably stands out as the most extensively employed for medicinal applications. This particular species has a rich history of traditional medicinal usage dating back to ancient times, with its prominence notably observed in the South and Southeast Asian regions. Indeed, its utilization in antiquity was frequently documented in Sanskrit medical literature and later found mention in Hindu and Buddhist historical records [9]. The extract derived from AN comprises a diverse array of constituents, including alkaloids, polyphenols, polysaccharides, and more, each of which exerts specific biological effects on the nervous system, digestive system, and circulatory system. Previous research has uncovered the notable impacts of ANs and ANEs in the realms of antidepressant, antibacterial, anti-inflammatory, and antioxidant activities. The main biomedical applications of ANE are summarized in Table 2.

### 4.1. Antidepressant Activity

Depression is a highly prevalent neuropsychiatric disorder with a substantial adverse impact on global public health. The etiology of depression is a multifaceted phenomenon encompassing diverse contributory elements, such as cerebrovascular damage, systemic inflammation, neurotransmitter dysregulation, and cerebral white matter lesions [63]. This neuropsychiatric illness is characterized by persistent symptoms of cognitive impairment and low mood, and in severe cases, individuals may experience suicidal tendencies. In individuals with depression, there are neuronal lesions in brain regions such as the hippocampus, prefrontal cortex, and amygdala, which result in reduced expression of neurotrophins, increased calcium influx, accumulation of intracellular calcium in hippocampal neurons, and generation of free radicals. These processes lead to neurotoxic effects and neuronal apoptosis. Dysregulation of the monoamines neurotransmitter system, including serotonin, norepinephrine, and dopamine, can also contribute to the development of depression [64]. Monoamine oxidase inhibitors (MAOIs) have shown clinical effectiveness in treating depressive disorders, particularly atypical depression. MAO-A is an enzyme that catalyzes the deamination of biogenic amines in the blood or synapses, thereby regulating the levels of dopamine, serotonin, norepinephrine, and catecholamine.

Ahsana Dar conducted a study that revealed the potential antidepressant activity of various fractions derived from AN, including the ethanol extract, hexane, ethyl acetate, and aqueous fractions. This was determined through classic antidepressant experiments conducted on rodents or rats [65,66]. Subsequently, Dar and colleagues reported that the dichloromethane fraction from AN exhibited antidepressant properties by inhibiting monoamine oxidase type A in rat brain homogenates [67]. The aqueous fraction of the AN was found to be the most potent inhibitor of MAO, with a similar effect to clorgyline, a specific MAO-A inhibitor. These findings suggest that AN contains separate compounds that contribute to its antidepressant and toxic effects. Nero et al. reported that arecoline, a compound found in AN, possesses the capacity to penetrate the blood–brain barrier and activate neuronal receptors. This mechanism of action leads to increased excitability in the body and has been associated with an antidepressant effect. Furthermore, studies have suggested that the alkaloids present in AN could potentially be utilized in the treatment of schizophrenia and depression. The efficacy of this treatment may be influenced by the consumption of AN [68].

Abbas et al. conducted a research investigation to explore the prospective antidepressant properties of ethanol extracts from ANs and their components in rats. They employed both acute and sub-chronic forced swimming tests to evaluate the effects, with IC_50_ values of 50 and 20 mg/kg, respectively. The researchers focused on the hippocampal area of the rats and compared the results to a control group. They found that the AN ethanol extract (50 mg/kg) and aqueous extract (20 mg/kg) significantly increased serotonin levels by approximately 35% and norepinephrine levels by approximately 30%, as determined by the high-performance liquid chromatography assay of monoamine levels. This increase in monoamine levels in the hippocampus of the rat brain is believed to be responsible for the observed antidepressant activity of the ANE and aqueous fraction [69]. The main active components responsible for this effect were identified as AN saponins, as supported by chemical analysis [70]. These saponins were found to increase serotonin and norepinephrine levels, consistent with the results of a prior investigation conducted by Alfaifi et al. Additionally, the researchers discovered that the dichloromethane fraction of the AN contains monoamines, primarily serotonin, with modulating potential. Importantly, this fraction did not produce a detectable cheese effect, similar to the effects of moclobemide (a MAO-A inhibitor). Therefore, the dichloromethane fraction holds potential as a viable candidate for the advancement of novel antidepressant drugs.

According to the monoamine hypothesis, brain monoamine levels, including serotonin, noradrenaline, and dopamine, serve as the foundational factor in depression. ANE is suggested to harbor potential antidepressant effects by elevating serotonin and noradrenaline levels [69,71]. The general mechanism is illustrated in Figure 3.

The research in molecular biology provides support for the development of the Mongolian medicine known as the Areca Thirteen Pill (GY-13) as a viable alternative for addressing depression. The findings indicate that the administration of GY-13 has a significant positive effect on depressive behavior, increases the levels of cAMP and PKA, restores the mRNA levels of CREB and BDNF, and enhances growth activity in the hippocampus. Additionally, through sucrose preference tests, open field tests, and body weight measurements conducted on rats, it has been observed that the effects of GY-13 treatment on CREB mRNA and BDNF mRNA levels can be counteracted by inhibiting PKA. This suggests that GY-13 treatment exhibits a strong antidepressant action by activating the cAMP/CREB/BDNF signaling pathway, promoting cell proliferation, and suppressing cell death [72].

### 4.2. Anti-Inflammatory Activity

Inflammation, an innate defense mechanism triggered in response to external stimuli, pathogens, and immune responses, play a pivotal role in the pathogenesis of numerous medical conditions, encompassing cardiovascular diseases, neurodegenerative disorders, diabetes, autoimmune maladies, and even tumors. Although nonsteroidal anti-inflammatory drugs (NSAIDs) have found widespread application in the management of inflammation-related ailments, their gastrointestinal and cardiovascular side effects consistently impose restrictions. Natural compounds emerge as a promising source for the development of anti-inflammatory agents owing to their structural diversity and recurrent bioactive properties [73]. Studies have demonstrated that the polyphenols in ANs have pharmacological actions that include analgesic, anti-inflammatory, anti-fatigue, antioxidant, and antibacterial qualities [74].

The acetone extract of ANs contains a significant amount of proanthocyanins, which effectively alleviate the edematous inflammation caused by carrageenan in rats and reduce the levels of prostaglandin E2. Additionally, the ethanol extract of ANs has been found to reduce foot swelling, with an inhibition rate of 86.79% after 24 h. The anti-inflammatory effect of ANs is comparable to that of aspirin, suggesting that ANEs may exert their anti-inflammatory effects through pathways involving the inhibition of cyclooxygenase (COX) or other anti-inflammatory mechanisms [75]. Furthermore, Huang et al. [76] discovered that the active components of ANs can lower the levels of interleukin-6 (IL-6) in tumor cells, increase the levels of the tumor suppressor factor p53, induce tumor cell apoptosis, and thus potentially prevent tumor development.

The primary anti-inflammatory mechanism of ANE is similar to that of most COX inhibitors. It operates mainly by inhibiting COX in the metabolism of arachidonic acid. This process leads to a reduction in the synthesis of prostaglandin (PG), thereby exerting an anti-inflammatory effect [75,77]. The general mechanism is illustrated in Figure 4.

Kweon et al. conducted a study in which they found that the water extract derived from the pericarpium of the areca nut (ARP) has the potential to alleviate chronic pancreatitis by deactivating pancreatic stellate cells. Administering ARP extract significantly improved glandular atrophy and inflammation in the pancreas, while also inhibiting collagen accumulation. The water extract from ARP exhibited the ability to impede the constituents of the extracellular matrix (ECM), specifically targeting alpha-smooth muscle actin (α-SMA), collagen I, and fibronectin 1 (FN1) in both pancreatic tissue and pancreatic stellate cells (PSCs) [78]. Furthermore, the water extract from AN demonstrated a notable anti-inflammatory effect on carrageenan-induced paw edema in mice and rats [75].

Chen et al. recently discovered eight novel phenethoxy derivatives, known as trichoasperellins A–H, which exhibit potent anti-inflammatory properties [79]. These compounds were isolated from the endophytic fungus *Trichoderma asperellum* G10 found in AN. In a separate study, it was observed that arecoline regulates the expression of genes involved in oxidative stress response in human keratinocytes [80]. Furthermore, Soo-Jin Jeong demonstrated that a traditional Korean herbal formula called Gwakhyangjeonggi-san (GHJGS), which contains Areca catechu, possesses both anti-inflammatory and antioxidant activities. This was confirmed through experiments involving the inhibition of LPS-induced phosphorylation of mitogen-activated protein kinases, increased expression of heme oxygenase-1, and prevention of reactive oxygen species generation in RAW 264.7 cells, as determined by Western blotting and enzyme-linked immunosorbent assay (ELISA) [81].

In a study conducted by Liu et al., it has been reported that arecoline has the potential to inhibit the proliferation of osteoclasts and stimulate the differentiation of osteoblasts, thus presenting a prospective therapeutic and prophylactic strategy for arthritis. The primary mechanism of action involves the interference of arecoline with signal pathways activated by AKT, MAPK, and NF-κB, leading to the suppression of gene expression and the translocation of genes associated with osteoclast differentiation. Conversely, arecoline can stimulate the expression of genes related to osteoblast differentiation, thereby facilitating joint cavity repair [82]. Furthermore, Yin et al. employed metabolomics analysis to demonstrate that arecoline possesses potent antioxidant properties and can enhance its anti-inflammatory effects by regulating reactive oxygen species (ROS) and hypoxia-inducible factor-1α (HIF-1α) [83].

Sun et al. employed an ultrasonic-assisted methodology to extract polyphenols from AN. As shown in Figure 5a, the effect of ACP on pre-osteoblasts was investigated by treating MC3T3-E1 cells with different ACP concentrations (25–150 μg/mL). Three significant markers of the differentiation of osteoblasts were detected, alkaline phosphatase (ALP) activity, the contents of collagen type I (COL-I) and osteocalcin (OCN) increased significantly, with consistent levels across ACP concentrations. In Figure 5b, the number of calcium nodules increased significantly in ACP-treated groups, showing a notable dose-dependent effect. Semi-quantitative analysis using cetylpyridinium chloride solution revealed substantial mineralization induction by ACP in MC3T3-E1 cells, with the highest increase (289.53%) observed at 150 μg/mL ACP dosage. Their research findings indicate that these polyphenols possess the capacity to enhance osteogenic proliferation, differentiation, and mineralization [84].

### 4.3. Antioxidant Activity

Reactive oxygen species (ROS) are continuously generated in the body due to oxidative biochemical reactions. Under normal conditions, there is a dynamic equilibrium between ROS production and scavenging. However, in specific circumstances, ROS production can become excessive, and our bodies may not effectively eliminate all ROS. Elevated levels of ROS can lead to cellular death and tissue damage, contributing to various human diseases, including aging, chronic arterial diseases, and even cancer. Currently, a wide array of artificially synthesized drugs are commonly employed to mitigate oxidative damage caused by ROS. However, accumulating evidence has illustrated that these chemical agents designed to inhibit ROS may carry potential health risks. Consequently, the exploration of innovative natural strategies to counteract surplus ROS is imperative.

Among natural antioxidants, phenolic compounds are considered the most important and widely available, serving as alternatives to synthetic antioxidants [85,86]. By replacing synthetic antioxidants, phenolic compounds can help reduce the harmful and carcinogenic effects of free radicals. ANs are particularly rich in natural antioxidants due to their high content of phenolic acids, tannins, and flavonoids. As the polyphenol content increases, the antioxidant effect becomes stronger. Polyphenols exert their antioxidant activity by neutralizing lipid free radicals or preventing the formation of free radicals from hydrogen peroxide. Additionally, research has shown that the original anthocyanins in ANs possess antioxidant capabilities, and the degree of polymerization of these anthocyanins is closely related to their antioxidant capacity [87]. Animal studies have also demonstrated that AN polyphenol can enhance antioxidant capacity and protect vital organs in rats exposed to acute high-altitude hypoxia. Furthermore, AN ethanol extract has been found to inhibit vascularization in plasma protein-nitroglycerin venous and local models, as well as inhibit the expression of inducible nitric oxide [88]. The antioxidant mechanism is shown in Figure 6.

The primary mechanism through which AN polyphenol exhibit anti-aging effects is by inhibiting elastase activity. A study conducted by Lee et al. [89] found that polyphenolic extracts derived from AN exhibited a protective effect against oxidative-stress-induced damage in V79-4 Chinese hamster lung fibroblast cells induced by H_2_O_2_. This protective effect was attributed to the elimination of free radicals and the enhancement of antioxidant enzyme activities, such as catalase (CAT), superoxide dismutase (SOD), and glutathione peroxidase (GPX). These polyphenols can protect connective tissue proteins by inhibiting elastase, leading to a significant reduction in skin tissue aging and inflammation. Furthermore, it was discovered that ANE-modified silver nanoparticles possessed antioxidant and antibacterial properties [90]. Another study confirmed the antioxidant properties of various extracts obtained from the AN, as evidenced by their lower IC_50_ values obtained through the FRAP assay compared to that of ascorbic acid [91]. Additional antioxidant tests, including the DPPH radical scavenging test, the H_2_O_2_ scavenging test, and the Fe^2+^ chelating test, demonstrated stronger antioxidant properties in the ethyl acetate extract and nonpolar solvent extracts, specifically n-hexane and chloroform. Overall, the antioxidant properties of the *Areca catechu* L. plant have been identified.

Hu et al. [92] conducted an optimization study on the extraction process of AN polysaccharides using the response surface methodology. They also investigated the antioxidant activity of these polysaccharides. The researchers determined that the optimal extraction conditions were a duration of 1.5 h, a solid-to-liquid ratio of 1:30 (g/mL), and two extraction cycles. The molecular weight of the AN polysaccharide ranged from approximately 1.84 × 10^4^ g/mol to 4.79 × 10^5^ g/mol. The polysaccharides were found to be composed of mannose, glucose, galactose, and arabinose.

In a separate study conducted by Zhang et al. [93], 11 phenolic compounds were isolated from the EtOAc extract of AN through bioassay-guided fractionation. These compounds exhibited significant antioxidant activity as determined by the DPPH radical scavenging assay. Another investigation evaluated the antioxidant activity of AN ethanol extract by measuring its DPPH radical scavenging activity and hydroxyl radical scavenging activity. The results indicated that the ethanol extract of AN, which contains epicatechin and syringic acid as the main polyphenols, displayed higher antioxidant activity compared to ascorbic acid (Vc). However, its antioxidant capacity was lower than that of Vc [94]. Furthermore, other studies successfully isolated and purified polysaccharides and polyphenols from AN. It was determined that one of these compounds exhibits antioxidant activity *in vitro* [95,96].

### 4.4. Antimicrobial Activity

Bacterial infections persistently threaten global human health. The overuse of antibiotics, a potent tool against bacterial pathogens, has directly contributed to the emergence of a grave concern: bacterial resistance. However, the antibacterial effect of existing antibacterial drugs is limited, and the research and development of new drugs are relatively slow. Therefore, there is an urgent imperative to explore and advance novel antibiotics. Natural products, with their wide sources and low side effects, represent a valuable resource for the development of antifungal drugs [97].

A study conducted by Jam et al. [98] investigated the antibacterial effects of ANEs using different solvents (methanol, ethanol, and water) against various bacteria including *Staphylococcus aureus*, *Escherichia coli*, *Salmonella enterica*, and *Enterobacter aerogenes* bacteria. The agar disc diffusion technique was employed to assess the antibacterial activity. The methanolic extract exhibited significant antibacterial activity, with a minimum inhibitory concentration value of 1.56 mg/mL against *Escherichia coli*. Furthermore, all extracts were tested for their antibacterial effects in a prepared cake, and it was observed that the methanolic extract at a concentration of 100 mg/L was the most effective against all pathogens, except for *Escherichia coli*, which showed greater sensitivity to the ethanolic extract. Moreover, AN polyphenols were found to inhibit the activity of *Mycobacterium tuberculosis* H37Rv [99]. The butanol extract components of ANs demonstrated the strongest antimalarial activity against *Plasmodium falciparum*, with an IC_50_ value of 18 μg/mL, indicating potent antiparasitic properties [100]. Additionally, Punnawich Yenjit confirmed that compounds extracted from the pericarp of ANs possess fungicidal activity against *Colletotrichum gloeosporioides* and have the potential to control postharvest mango anthracnose both *in vitro* and in mango fruit [101].

The antimicrobial properties of betel nut primarily function through two mechanisms. The first involves a direct impact on microorganisms, disrupting their cell walls and resulting in microbial death [102,103]. The second enhances the host’s immune capacity by activating leukocyte to effectively clear microorganisms [104]. The general mechanism is illustrated in Figure 7.

**Figure 7 pharmaceuticals-17-00228-f007:**
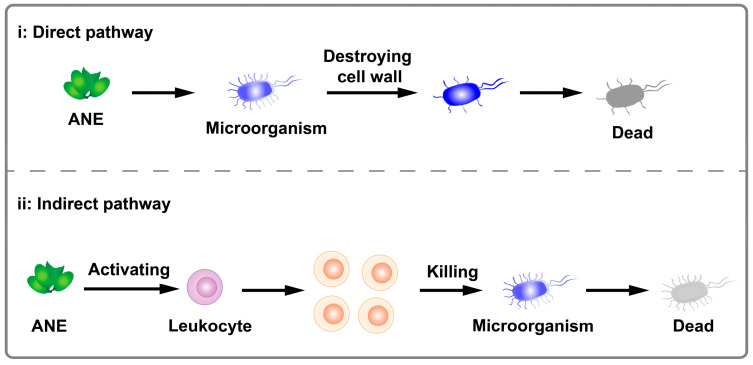
Schematic illustration of antimicrobial activity of ANE.

**Table 2 pharmaceuticals-17-00228-t002:** A summary of the pharmacological effects and details of both ANs and ANEs.

Pharmacological Effects	Main Bioactive Phytoconstituents	Main Experimental Methods	Dosage	Main Mechanism of Action	Reference
Antidepressant activity	Ethanol, hexane, ethyl acetate, aqueous extract fractions of the AN	Forced swim test; tail suspension test; locomotion test; Monoamine oxidase assay.	2.5, 5, 10, 13, 20 and 50 mg/kg	The alkaloids viz arecaine, arecaidine, and arecoline have been shown to be present in the AN with only arecoline having MAO-inhibiting properties. The possibility that the MAO inhibition in the present report may be due to arecoline and some other unknown compounds needs further investigation.	[66]
	Dichloromethane and aqueous extract of the AN	Tail-suspension test; yohimbine potentiation test; locomotor test; monoamine oxidase assay.	5 mL/kg and 10 mL/kg for rats and mice	Inhibiting monoamine oxidase type A in rat brain homogenates.	[67]
	Ethanol extracts from AN	Acute and sub-chronic forced swimming tests.	AN ethanol extract (50 mg/kg), aqueous extract (20 mg/kg)	Increased serotonin levels by approximately 35% and norepinephrine levels by approximately 30%.	[69]
	Areca Thirteen Pill (GY-13)	Sucrose preference tests, open field tests, and body weight measurements.	0.25 g/kg (low), 0.50 g/kg (medium), 1.00 g/kg (high)	Increases the levels of cAMP and PKA, restores the mRNA levels of CREB and BDNF, and enhances growth activity in the hippocampus.	[72]
Anti-inflammatory activity	Acetone, ethanol extract of ANs	Paw edema in rats; free radical scavenging activity.	10, 50, and 100 mg/kg	Carrageenan in rats and reduce the levels of prostaglandin E2; the inhibition of cyclooxygenase.	[75]
	Active components of ANs	Quantitative real-time PCR analysis; detection of IL-6 production; apoptosis analysis; cell cycle analysis.	2 mg/kg	Lower the levels of interleukin-6 (IL-6); increase the levels of the tumor suppressor factor p53	[76]
	The water extract derived from the pericarpium of AN	Biochemical analysis; Masson’s trichrome staining Immunofluorescence staining; real-time RT-PCR; Western blot.	50, 100, or 200 mg/kg	Deactivating pancreatic stellate cells; the water extract from ARP exhibited the ability to impede the constituents of the extracellular matrix (ECM), specifically targeting alpha-smooth muscle actin (α-SMA), collagen I, and fibronectin 1 (FN1) in both pancreatic tissue and pancreatic stellate cells (PSCs).	[78]
Anti-inflammatory activity	Gwakhyangjeonggi-san (GHJGS)	LPS-induced phosphorylation of mitogen-activated protein kinases; Western blotting and enzyme-linked immunosorbent assay (ELISA).	0, 250, 500 or 1000 µg/mL	Increased expression of heme oxygenase-1 and prevention of reactive oxygen species generation.	[81]
Anti-inflammatory activity	Arecoline	Bone marrow macrophage (BMM) isolation; cell viability assay; osteoclast differentiation; tartrate-resistant acid phosphatase (TRAP) staining; real-time RT-PCR; Western blot.	10 mg/kg	The interference of arecoline with signal pathways activated by AKT, MAPK, and NF-κB, leading to the suppression of gene expression and the translocation of genes associated with osteoclast differentiation.	[82]
	AN polyphenols	Cell viability assays; specific marker protein detection; fluorescence investigations	0, 25, 50, 100, 150 μg/mL	Areca nut seed polyphenol (ACP) remarkably enhanced the content of ALP in osteoblasts, which promoted the differentiation of osteoblasts.	[84]
Antioxidant activity	AN polyphenol; ethanol extract from AN	Quantitative real-time PCR analysis; Western blot; free radical scavenging activity assays.	Oral administration of ANE (250 mg/kg and 500 mg/kg)	Inhibit vascularization in plasma protein-nitroglycerin venous and local models, as well as inhibit the expression of inducible nitric oxide.	[88]
	AN polyphenol	Quantitative real-time PCR analysis; Western blot; free radical scavenging activity assays.	IC_50_ values were 26.9 mg/mL for porcine pancreatic elastase and 60.8 mg/mL for human neutrophil elastase.	The elimination of free radicals and the enhancement of antioxidant enzyme activities, such as catalase (CAT), superoxide dismutase (SOD), and glutathione peroxidase (GPX).	[89]
	AN polyphenol	Free radical scavenging activity assays.	IC_50_ = 1483.43 μg DW/mL	Inhibiting elastase, reduction in skin tissue aging and inflammation.	[90]
	EtOAc extract from AN	DPPH radical scavenging assay; hydroxyl radical scavenging activity.	20, 40, 60, 80, 120, and 200 µg/mL	The polyphenols in the ethanol extract of areca nut seed can remove free radicals and exert antioxidant activity.	[94]
Antimicrobial Activity	Methanol, ethanol, and water extract from ANs	Agar disk diffusion assay; microtiter broth dilution method; agar disc diffusion technique.	The agar disc diffusion technique: 100 mg/L	AN is rich in tannins, terpenoids, alkaloids, and flavonoids, and tannin in AN is effective to inhibit gram-positive bacteria. Tannin binds with peptide on the peptidoglycan component from cell walls, which in turn disturb the integrity of bacterial cell walls, which cause bacterial cell damage. Finally, it led to disturbance of the metabolism process and, subsequently, the death of cells.	[98]
	AN polyphenols	Microwave and Soxlet apparatus; resazurin microtiter assay.	MIC: 0.975 ± 0.02 µg/mL	The bioactivity of the extract was attributed to the nontoxic polyphenols present. This extract also showed selective inhibition of M. tuberculosis over other gram-positive and gram-negative bacteria.	[99]
	ANE	High-performance liquid chromatography (HPLC) method; phagocytosis assay; analysis of macrophage activity and capacity; analysis of serum biochemistry.	Oral administration of ANE (500, 1000, or 1500 mg/kg)	Administration of areca nut extract increased the number of WBCs and improved the activity and capacity of macrophages significantly in rats infected with *S. aureus*.	[102]

A recent scientific study utilized scanning electron microscopy (SEM) images to examine the impact of ANE-modified silver nanoparticles (AgNPs) on bacterial cells. The results indicated that clusters of AgNPs were observed adhering to the surface of the bacterial cell walls, suggesting potential harm to the cell membranes [105]. This composite nanoparticle has been increasingly utilized as a promising and efficient antibacterial agent in recent years [106]. The addition of areca leaf extract to chitosan and chitosan/vanillin (CH/Vn) blend films has been shown to enhance the antibacterial activity of chitosan films [107]. Additionally, the inclusion of ANE in the soap production process has the potential to impact the quality characteristics and sensory properties of glycerine bar soap, while also enhancing its antibacterial activity [89]. Additionally, in an experimental setting involving rats infected with *Staphylococcus aureus*, the administration of ANE resulted in a notable increase in white blood cell count and a significant enhancement in the activity and capability of macrophages, thus enhancing the antibacterial ability of rats. This immunomodulatory effect is likely attributed to the presence of a phenolic compound within the AN [102].

In a study conducted by Fan et al. [108], the impact of various extraction methods on the antibacterial and antioxidant properties of phenolic compounds found in AN was assessed. The results indicated that the MIC (minimum inhibitory concentration) values for *Streptococcus mutans*, *Staphylococcus aureus*, *Escherichia coli*, and *Pseudomonas aeruginosa* were 2.50, 5.00, 15.00, and 25.00 mg/mL, respectively. These findings suggest a potential correlation between the antibacterial and antioxidant activities of AN and the presence of proanthocyanidin polymers.

## 5. Conclusions

Researchers have elucidated its biological effects on the nervous system, digestive system, and blood circulatory system, exploring the potential of AN and ANEs in various roles, including as antidepressants, anti-inflammatory agents, antioxidants, and antibacterial agents. They have also revealed its mechanism of action through a series of techniques. During the antidepressant process, ANE elicits its antidepressant effects through a variety of mechanisms, including its capacity to traverse the blood–brain barrier and activate neuronal receptors. Additionally, it accomplishes this by inhibiting monoamine oxidase type A, increasing serotonin and norepinephrine levels, activating the signaling pathway of cAMP/CREB/BDNF, promoting cell proliferation, and suppressing cell death. Consequently, it contributes to the improvement of impaired cerebral vasculature, systemic inflammation, perturbed neurotransmitter levels, and pathological white matter lesions, thereby exerting an antidepressant effect. In the anti-inflammatory process, ANE inhibits cyclooxygenase or interferes with the pro-inflammatory signal pathways activated by AKT, MAPK, and NF-κB, or reduces inflammation-related cytokines (IL-1, IL-6, G-CSF, and NF-κB and GM-CSF), inhibits LPS-induced phosphorylation of mitogen-activated protein kinases, and increases the expression of heme oxygenase-1 and the prevention of reactive oxygen species generation. In the antioxidant process, ANE (mainly polyphenols) scavenges free radicals generated by oxidation and enhances the activity of antioxidant enzymes, thereby fulfilling the function of an antioxidant. In the antibacterial process, ANEs (mainly tannins) can interact with the bacterial cell wall to disrupt its integrity or regulate the bacterial metabolism, thus performing an antibacterial function.

AN, as a natural product, is abundant in production and widely available. Although it is extensively used in traditional Chinese medicine and clinics. Researchers have elucidated its biological effects on the nervous system, digestive system, and blood circulatory system, exploring the potential of ANs and ANEs in various roles, including as antidepressants, anti-inflammatory agents, antioxidants, and antibacterial agents. The future prospects of ANs and ANEs as pharmaceutical or pharmaceutical adjunct components hold promise for potential medicinal applications. Ongoing research into the bioactive compounds present in AN provides opportunities for the development of novel drugs. However, it is still necessary to conduct more in-depth research on its biological effects and develop a comprehensive understanding of its interaction with the various cells, tissues, and organs of the living body, clarifying its adverse components, mechanisms of action, and pharmacology and, then, reasonably designing new drugs with high efficiency and low toxicity, so as to provide scientific guidance for human health and disease prevention and treatment.

## Figures and Tables

**Figure 1 pharmaceuticals-17-00228-f001:**
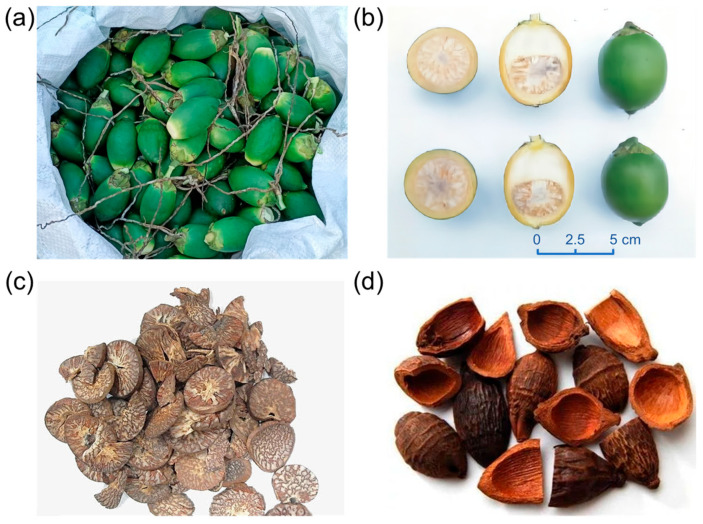
Areca nuts and their use in traditional Chinese medicines. (**a**) Fresh betel nuts. (**b**) Transverse and longitudinal section views of fresh betel nut [2], Copyright 2021 Wiley-VCH. (**c**) Sliced kernels of dried betel nut (semen arecae); longitudinal cut image of fresh betel nut fruit that has lost a certain amount of moisture. (**d**) Dry, peeled areca pericarp.

**Figure 2 pharmaceuticals-17-00228-f002:**
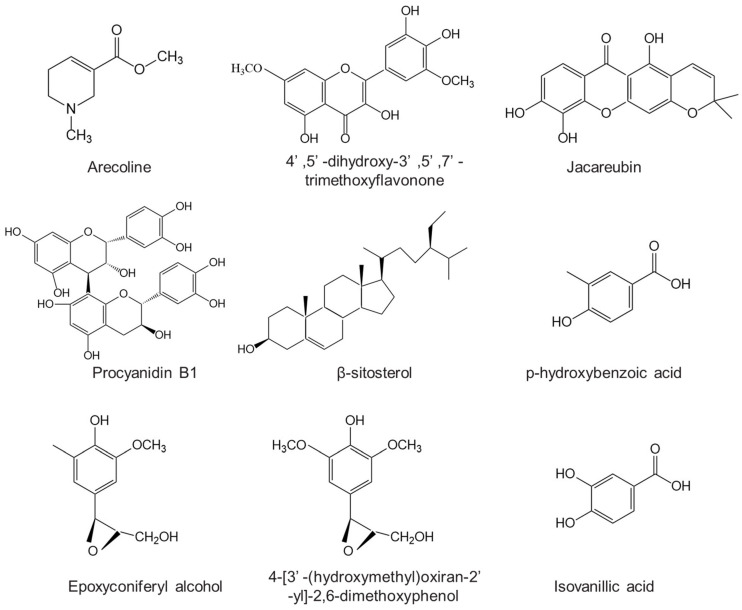
The main bioactive phytoconstituents of areca nut.

**Figure 3 pharmaceuticals-17-00228-f003:**
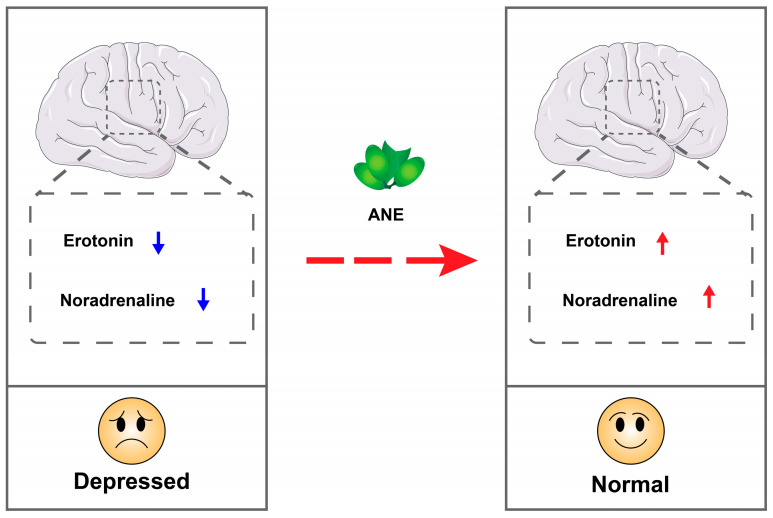
Schematic illustration of antidepressant activity of ANE.

**Figure 4 pharmaceuticals-17-00228-f004:**
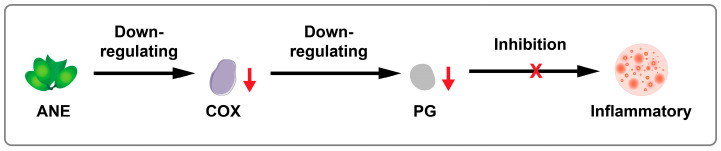
Schematic illustration of anti-inflammatory activity of ANE.

**Figure 5 pharmaceuticals-17-00228-f005:**
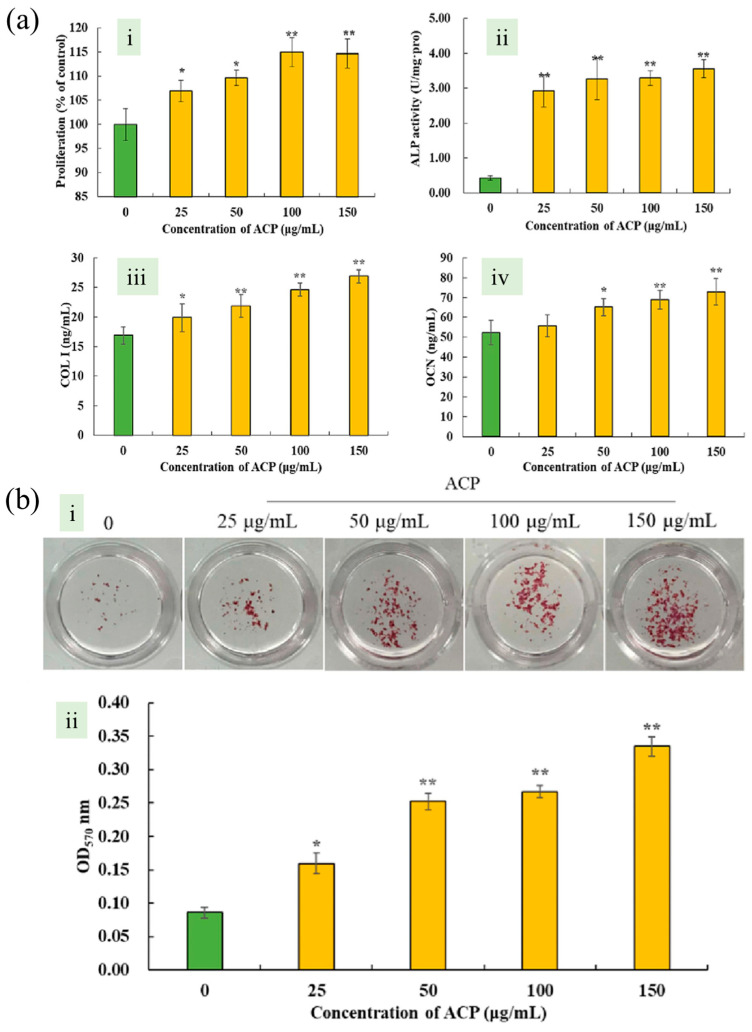
Areca nut seed polyphenol (ACP) induce the proliferation, differentiation and mineralization of osteoblasts *in vitro*. (**a**) (i) Effect of areca nut seed polyphenol (ACP) on proliferation, (ii) alkaline phosphatase (ALP) activity, (iii) collagen type I (COL-I) content, (iv) osteocalcin (OCN) content of pre-osteoblast MC3T3-E1. (**b**) (i) Effect of ACP on mineralized nodules of pre-osteoblast MC3T3-E1, Image of MC3T3-E1 stained with 1% Alizarin Red S solution. (ii) Graph showing semi-quantitative experimental results. [84]. Copyright 2021 American Chemical Society. * *p* < 0.05, ** *p* < 0.01.

**Figure 6 pharmaceuticals-17-00228-f006:**
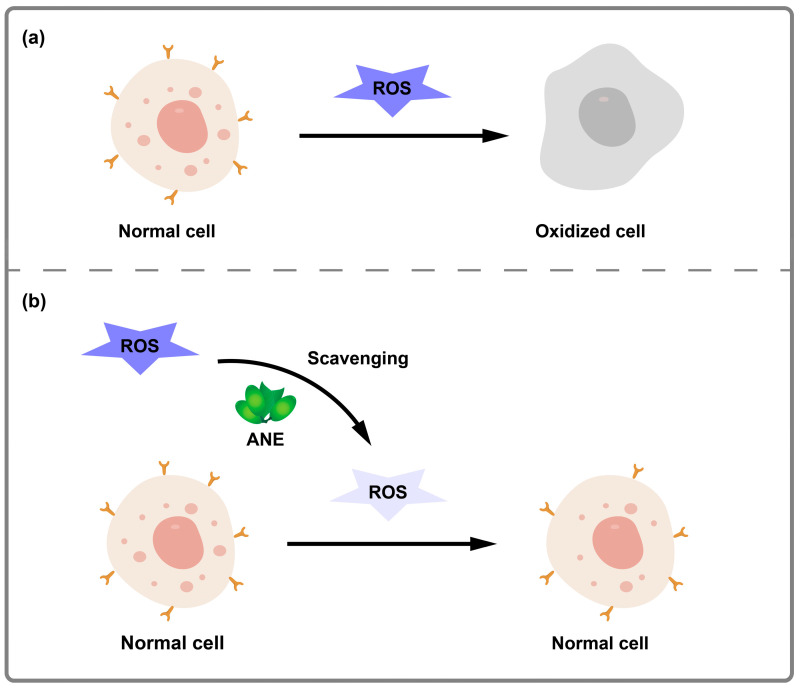
Schematic illustration of antioxidant activity of ANE. (**a**) The process of normal cells oxidized by a high level of ROS. (**b**) The process of normal cells oxidized with a low level of ROS scavenged by ANE.

**Table 1 pharmaceuticals-17-00228-t001:** Chemical compounds isolated from AN.

Classification	Chemical Component	Reference
Alkaloids	Arecoline, Arecaidine, Guavacoline, Guavacine, Arecolidine, Ethyl N-methyl-l,2,5,6-tetrahydro-pyridine-3-carboxylate, Methyl nicotinate, Ethyl nicotinate, Methyl N-methylpiperidine-3-carboxylate, Ethyl N-methylpiperidine-3-carboxylate, Nicotine, Isoguvacine, Homoarecoline	[22]
Flavonoids	Isorhamnetin, Chrysoeriol, Luteolin, Quercetin, 4′,5′-dihydroxy-3′,5′,7′-trimethoxyflavonone, 5,7,4′-trihydroxy-3′,5′-di methoxy flavanone, Liquiritigenin, Jacareubin,	[1,23]
Tannins	Catechin, Epicatechin, Procyanidin A1, Procyanidin B1, Procyanidin B2, Arecatannin A1, Arecatannin B1, Arecatannin C1, Arecatannin A2, Arecatannin A3, Arecatannin B2	[24,25]
Triterpenoids and steroidals	Arborinol, Arborinol methyl ether, Ursonic acid, 3β-acetyl ursolic acid, Cycloartenol, Fernenol, Arundoin, 5,8-epidioxiergosta-6,22-dien-3β-ol, Stigmasta-4-en-3-one, β-sitosterol	[26]
Fatty acids	Lauric acid, Myristic acid, Palmitic acid, Stearic acid, Oleic acid	[1]
others	Chrysophanol, Physcion, Resveratrol, Ferulic acid, p-hydroxybenzoic acid, Vanillic acid, de-O-methyllasiodiplodin, Epoxyconiferyl alcohol, 4-[3′-(hydroxymethyl) oxiran-2′-yl]-2,6-dimethoxyphenol, Protocatechuic acid, Isovanillic acid, Cyclo-(Leu-Tyr)	[1]

## Data Availability

Data sharing is not applicable.

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
