# Peer review of "Biological Effects and Biomedical Applications of Areca Nut and Its Extract"

_pharmaceuticals, 2024, doi:10.3390/ph17020228_

Round 1
Reviewer 1 Report
Comments and Suggestions for Authors
The manuscript entitled 'Biological effects and biomedical applications of areca nut and its extract' needs a major revision.
1. A moderate check is required for the English language and style through the text.
2. I propose to add the terms 'toxicity' and 'antidepressant activity' to the list of keywords.
3. Table 1 'The chemical composition of areca nuts' should be improved and widened by processing more sources. The necessary references should be added to each line of described compounds. The terms in the first column should be used in plural (Alkaloids, tannins, etc.)
4. For better visualization I highly recommend to the authors to prepare Figure 2 with the formulas of their main bioactive phytoconstituents.
5. The italic type should be used everywhere for writing Latin names of species and for terms in vitro, in vivo for instance - Lines 100, 192, 209 (Piper betle), etc.
6. Since the term 'Areca nut' occurs more than 100 times in the text, I will suggest using an abbreviation to denote it.
7. I highly recommend the s authors also create a Table 2 in which data from scientific sources with experimental (indicating the model used) and clinical studies regarding the pharmacological activity of areca nuts and their extracts would be presented (for a better generalization and visualization of the analyzed literary sources).
8. The Chapter 5. Conclusions should be shortened and presented more succinctly. In addition, it should not contain references to literary sources.
9. Some aspects of the toxicity of the areca nuts should be discussed more comprehensively (chapters 3 or 4). There are a lot of publications in the PubMed database regarding this aspect, for instance –
· Mittal SS, Mohod S, Lohe V, Patel S, Khubchandani M, Kukde MM. Role of Betel Nut in Liver Toxicity in Oral Submucous Fibrosis and Oral Cancer Patients: A Case-Control Study. Cureus. 2023 Nov 9;15(11):e48562. doi: 10.7759/cureus.48562. PMID: 38073941; PMCID: PMC10710118.
· Mahdavi Mortazavi SM, Ataollahi M, Mashhadiagha A, Moosavi SA, Moshfeghinia R, Soheili M. Acute toxic effects of areca nut on central nervous system and liver: A case report. Clin Case Rep. 2023 Oct 4;11(10):e7976. doi: 10.1002/ccr3.7976.
· Kong D, Wang G, Tang Y, Guo M, Ul Haq Khan Z, Guo Y, Gu W, Ma Y, Sui M, Li J, Yang M. Potential health risk of areca nut consumption: Hazardous effect of toxic alkaloids and aflatoxins on human digestive system. Food Res Int. 2022 Dec;162(Pt A):112012. doi: 10.1016/j.foodres.2022.112012.
· Myers AL. Metabolism of the areca alkaloids - toxic and psychoactive constituents of the areca (betel) nut. Drug Metab Rev. 2022 Nov;54(4):343-360. doi: 10.1080/03602532.2022.2075010
Comments on the Quality of English LanguageA moderate check is required for the English language and style through the text. For instance, in the Abstract I have marked some issues: The dried, mature fruit of the palm tree species Areca catechu L. (Arecaceae) is known as the areca nut, or betel nut. It is widely cultivated in the tropical regions. In many nations, areca nuts are utilized for traditional herbal treatments or social activities. Areca nuts have historically been used to address various health issues, such as diarrhea, arthritis, dyspepsia, malaria and so on. In this review, we have conducted a comprehensive summary of the biological effects and biomedical applications as well as some toxicity aspects of areca nuts and their extracts. Initially, we provided an overview of the constituents in areca nuts' extracts. Subsequently, we summarized the biological effects of areca nuts and their extracts on the digestive ystem, nervous system, and circulatory systems. Subsequently, we elucidated the contributions of areca nuts and their extracts in the applications of antidepressant, anti-inflammatory, antioxidant, and antibacterial. Finally, we discuss the challenges and future perspectives regarding the utilization of areca nuts and phytosubstances from them as emerging pharmaceuticals or valuable adjuncts within the pharmaceutical field.
Reviewer 2 Report
Comments and Suggestions for Authors
The review article titled " Biological effects and biomedical applications of areca nut and 2 its extract" is focused on the Areca catechu species
The abstract : The dried, mature fruit of the palm tree species Areca catechu L. is known as the areca nut, or betel nut. It is widely cultivated in the tropical regions. In many nations, areca nut is utilized for 10 traditional herbal treatments or social activities. Areca nut has historically been used to address 11 various health issues, such as diarrhea, arthritis, dyspepsia, malaria and so on. In this review, we have conducted a comprehensive summary of the biological effects and biomedical applications of areca nut and its extracts. Initially, we provided an overview of the constituents in areca nut extract. Subsequently, we summarized the biological effects of areca nut and its extracts on the digestive system, nervous system, and circulatory system. Subsequently, we elucidated the contributions of areca nut and its extracts in the applications of antidepressant, anti-inflammatory, antioxidant, and antibacterial. Finally, we discuss the challenges and future perspectives regarding the utilization of areca nut and its extracts as emerging pharmaceuticals or valuable adjuncts within the pharmaceutical field.
Tha abstract is well written.
Regarding the introduction: Authors should include the references in Table 1. Mentioning each chemical compound found in the literature.
Regarding the biomedical application explain other uses related with anticancer properties. Furthermore, addtional graphical must be included in each activity explaining the biological effect.
Reviewer 3 Report
Comments and Suggestions for Authors
This manuscript explores the medicinal applications of Areca catechu L. (areca nut), emphasizing its traditional uses, biological effects, and potential biomedical applications. It discusses the chemical composition, effects on the digestive, nervous, and circulatory systems, and highlights antidepressant properties. The review concludes by addressing challenges and prospects for areca nut as an emerging drug. Following points need careful attention of the authors when revising the manuscript:
How do specific compounds in areca nut contribute to its antidepressant effects?
What challenges exist in translating these findings into clinical applications?
I suggest the authors elaborate on potential side effects or safety considerations associated with areca nut use?
Reviewer 4 Report
Comments and Suggestions for Authors
The paper entitled “Biological effects and biomedical applications of areca nut and its extract” is a review article. Consequently, it is important to contain the chemical formulas of the major, at least, metabolites of every class, as are presented in Table 1. Emphasis should be given on the activity of isolated compounds and not only the extracts of the plant.
In Table 1 references are also needed for every class of active compounds.
Moreover, it is necessary to add a discussion and references concerning the correlation of the secondary metabolites on the biological activities of areca nut and not only for the case of arecoline.
Page 6, lines 235-237, a reference is missing.
Page 8, lines 311-313 and Page 8, lines 345-347 references are missing.
Page 9, line 365, Reference 67 concerns another plant (Piper betle) and no Areca nut.
Reviewer 5 Report
Comments and Suggestions for Authors
This review focuses on the bioactivity of the extract and its constituent, mainly arecoline, from areca nuts. The text is well-written and provides comprehensive information on the biological activity of areca nuts. However, the following points should be considered before the final decision.
Major point
I). There are several reviews on the toxicity of areca nut, such as,
1. The Controversial Roles of Areca Nut: Medicine or Toxin?
Liu, Pei-Feng ; Chang, Yung-Fu
International Journal of Molecular Sciences (2023), 24(10), 8996
2. Areca catechu L. (Arecaceae): A review of its traditional uses, botany, phytochemistry, pharmacology and toxicology
Peng, Wei; Liu, Yu-Jie; Wu, Na; Sun, Tao; He, Xiao-Yan; Gao, Yong-Xiang; Wu, Chun-Jie
Journal of Ethnopharmacology (2015), 164, 340-356
3. A review of the systemic adverse effects of areca nut or betel nut.
Garg, Apurva; Chaturvedi, Pankaj; Gupta, Prakash C
Indian journal of medical and paediatric oncology : official journal of Indian Society of Medical & Paediatric Oncology (2014), 35(1), 3-9
4. Chemical Composition of Areca Nut and Its Adverse Effects on Human Health.
Dangore-Khasbage, Suwarna; Bhowate, Rahul R; Khubchandani, Monika
Cureus (2023), 15(8), e43739
5. Carcinogenic Effects of Areca Nut and Its Metabolites: A Review of the Experimental Evidence.
Senevirathna, Kalpani ; Pradeep, Roshan ; Jayasinghe, Yovanthi Anurangi; Jayawickrama, Shalindu Malshan ; Illeperuma, Rasika ; Warnakulasuriya, Saman ; Jayasinghe, Ruwan Duminda
Clinics and practice (2023), 13(2), 326-346
6. A review on harmful effects of pan masala.
Garg, Apurva; Chaturvedi, P; Mishra, A; Datta, S
Indian journal of cancer (2015), 52(4), 663-6
7. Toxic effects of betel quid
Adhikari, Aniket; De, Madhusnata
International Journal of Human Genetics (2013), 13(1), 7-14
Please discuss the potential adverse effects to avoid the non-regulated use of areca nuts regarding general health concerns.
II) Please include the previously published reviews on biological activity and indicate the difference from past reviews.
1. A phytopharmacological review on therapeutic potential of areca catechu
Triveni, Girammagari; Girish, C.
Indo American Journal of Pharmaceutical Sciences (2020), 7(4), 262-266
2. Areca catechu L. (Arecaceae): A review of its traditional uses, botany, phytochemistry, pharmacology and toxicology
Peng, Wei; Liu, Yu-Jie; Wu, Na; Sun, Tao; He, Xiao-Yan; Gao, Yong-Xiang; Wu, Chun-Jie
Minor points
1. In Table 1, there are many mistakes.
luteolin: duplicated
glycyrrhizin: not Flavonoid. Is it isolated from areca nuts? It is a constituent of Glycyrrhiza glabra.
jacareubin: not flavonoid
zephyrin: no record
Please check the following triterpenoid compounds again. Some may be copy and paste problems, romatic alcohol, aromatic methyl ether, aroin, and cyclic altonil.
Please check the following fatty acids again, nutmeg acid, hexadanic acid.
Mannose, glucose, galactose, and arabinose are monosaccharides, not polysaccharides.
2. On page 7 line 301, “Abbas et al. a research investigation to explore.....” Please compose the sentence correctly.
3. Please use italic format for the scientific name, such as us “trichoderma asperellum” on page 9 line 368, and “Areca catechu” on page 10 line 441.
Comments on the Quality of English LanguageMinor revision to the main text is indicated in the comment.
Round 2
Reviewer 1 Report
Comments and Suggestions for Authors
lines 383, 575- the term 'cyclooxygenase' should not be used but only its abbreviation COX (because it was firstly abbreviated in line 377)
Reviewer 5 Report
Comments and Suggestions for Authors
The revised version is acceptable.
Comments on the Quality of English LanguageThe English is well-written. The editorial process may provide minor corrections.